# Genetic Testing and Surveillance of Young Breast Cancer Survivors and Blood Relatives: A Cluster Randomized Trial

**DOI:** 10.3390/cancers12092526

**Published:** 2020-09-05

**Authors:** Maria C. Katapodi, Chang Ming, Laurel L. Northouse, Sonia A. Duffy, Debra Duquette, Kari E. Mendelsohn-Victor, Kara J. Milliron, Sofia D. Merajver, Ivo D. Dinov, Nancy K. Janz

**Affiliations:** 1Department of Clinical Research, Faculty of Medicine, University of Basel, 4055 Basel, Switzerland; chang.ming@unibas.ch; 2School of Nursing, University of Michigan, Ann Arbor, MI 48109-5482, USA; lnortho@umich.edu (L.L.N.); karimend@umich.edu (K.E.M.-V.); 3College of Nursing, Ohio State University, Columbus, OH 43210, USA; duffy.278@osu.edu; 4Feinberg School of Medicine, Northwestern University, Chicago, IL 60611, USA; debra.duquette@northwestern.edu; 5Comprehensive Cancer Center, University of Michigan, Ann Arbor, MI 48109-5618, USA; kmilliro@med.umich.edu; 6School of Public Health, University of Michigan, Ann Arbor, MI 48109-5618, USA; smerajve@umich.edu (S.D.M.); nkjanz@umich.edu (N.K.J.); 7Statistics Online Computational Resource, School of Nursing, University of Michigan, Ann Arbor, MI 48109-2003, USA; dinov@umich.edu

**Keywords:** HBOC, statewide random sampling, cancer survivorship, targeted intervention, tailored intervention, Black participants, family recruitment, cascade genetic testing in families

## Abstract

**Simple Summary:**

Identifying breast cancer patients with pathogenic mutations that run in their families may improve the follow-up care they receive and breast cancer screening of their close relatives. In this study we identified breast cancer patients with high chances of having a pathogenic mutation and their close female relatives. We developed and tested two different kinds of letters and booklets that presented either personalized or generic information about screening and breast cancer that runs in families, and we encouraged participants to seek genetic evaluation. We found that both types of letters worked equally well for breast cancer patients and for relatives, regardless of their racial background. The personalized letters had slightly better outcomes. Some breast cancer patients and their relatives used genetic services and improved their screening practices. Black patients and their relatives were more satisfied with the booklets than other participants.

**Abstract:**

We compared a tailored and a targeted intervention designed to increase genetic testing, clinical breast exam (CBE), and mammography in young breast cancer survivors (YBCS) (diagnosed <45 years old) and their blood relatives. A two-arm cluster randomized trial recruited a random sample of YBCS from the Michigan cancer registry and up to two of their blood relatives. Participants were stratified according to race and randomly assigned as family units to the tailored (*n* = 637) or the targeted (*n* = 595) intervention. Approximately 40% of participants were Black. Based on intention-to-treat analyses, YBCS in the tailored arm reported higher self-efficacy for genetic services (*p* = 0.0205) at 8-months follow-up. Genetic testing increased approximately 5% for YBCS in the tailored and the targeted arm (*p* ≤ 0.001; *p* < 0.001) and for Black and White/Other YBCS (*p* < 0.001; *p* < 0.001). CBEs and mammograms increased significantly in both arms, 5% for YBCS and 10% for relatives and were similar for Blacks and White/Others. YBCS and relatives needing less support from providers reported significantly higher self-efficacy and intention for genetic testing and surveillance. Black participants reported significantly higher satisfaction and acceptability. Effects of these two low-resource interventions were comparable to previous studies. Materials are suitable for Black women at risk for hereditary breast/ovarian cancer (HBOC).

## 1. Introduction

Women diagnosed with breast cancer younger than 45 years old (young breast cancer survivors-YBCS) constitute approximately 25% of new breast cancer cases in the US, and are more likely to carry germline pathogenic variants associated with hereditary breast and ovarian cancer (HBOC) syndrome [1,2]. National guidelines recommend periodic screening for changes in family history and genetic evaluation (counseling and testing) of YBCS to determine HBOC status, and physical exams including clinical breast exams (CBE) and mammograms to screen for local recurrence or a new primary tumor [3,4]. However, there is underutilization of cancer genetic services and mammography surveillance among YBCS [5,6,7,8], especially among Black women [9,10,11,12], primarily due to lack of physician referral of minority women to genetic services, and due to complex barriers related to low income and low educational attainment that influence accessibility and acceptability of genetic services [13,14]. First- and second-degree relatives of YBCS have a 2.3 and 1.5 increased relative breast cancer risk respectively [15]. Relatives of HBOC cases should initiate MRI screening at age 25 (MRI) or earlier if indicated based on family history [3,4]. However, they may not always manage this risk effectively due to lack of information and inaccurate understanding of cancer risk inheritance patterns [16,17,18].

Paired with physician recommendations, theory-based and tailored interventions promote repeat screening, especially among racially diverse women [19,20,21,22,23,24,25]. However, the challenges concerning YBCS and at-risk relatives are firstly, the ability to identify them in large numbers, including racially diverse samples, and secondly, identifying low-resource ways to deliver information about the need for genetic evaluation and cancer surveillance guidelines. Two previous randomized controlled trials (RCTs) have shown that genetic counseling delivered over the telephone was not inferior to in-person counseling for cancer patients and at-risk relatives, yielded cost savings, and was equally acceptable to in-person counseling for disseminating information about screening guidelines [26,27]. A third RCT compared the efficacy of telephone genetic counseling versus a brochure and reported that about 40% of the telephone and 5% of the brochure groups obtained genetic counseling during the study period [28]. However, these studies included less than 10% of Black women and required significant resources for delivering the intervention and recruiting high-risk participants.

The present RCT builds on this prior work by oversampling for Black participants and by comparing the efficacy of two low-resource interventions delivered via postal mail, which included a targeted (more generic) versus a tailored (person-specific) intervention. The outcomes presented in this paper are initiation of genetic testing for YBCS, cascade genetic testing for relatives, and surveillance (CBE and mammography screening) consistent with national guidelines for YBCS and relatives. Satisfaction, acceptance, and perceived usefulness of the interventions were also assessed.

### Interventions

The Theory of Planned Behavior (TPB) [29] guided the development of the targeted and the tailored intervention. Table 1 presents how the components of each intervention correspond to the constructs of the TPB. The main message in both interventions was that early age of breast cancer onset is a “red flag” for hereditary disease and that participants should seek genetic evaluation. The targeted intervention included a letter and a booklet written at a seventh-grade reading level, which provided information about genetic counseling, cost, a list of certified cancer genetic services in Michigan, and online genetic resources. The booklet presented mammography screening as more sensitive than CBE and more accessible compared to MRI [30] and options for low cost screening. The targeted letter also included National Comprehensive Cancer Network (NCCN) guidelines for follow-up care (YBCS) and screening (relatives), and recommended that they seek genetic evaluation and breast surveillance/screening due to their own or their relative’s early age of cancer onset.

The tailored intervention included the same booklet as above and a second booklet presenting basic principles of open communication and family support. The purpose of this second booklet was to enhance the tailored messages by encouraging participants to maintain open communication for family challenges associated with early cancer onset, and mobilize family support for obtaining genetic services and surveillance. Based on participants’ responses to the baseline survey, a computer algorithm generated a letter, which provided tailored feedback about the need to have genetic evaluation and surveillance/screening. Two messages were generated for dichotomous tailoring variables, i.e., had genetic testing (yes/no) and frequency of surveillance consistent with guidelines (yes/no). Two messages were also generated for continuous variables. Self-efficacy and intention for genetic testing and surveillance were scored from 1 to 7 and a cut-off score of ≤3.5 was used to identify participants with low versus high self-efficacy and intention. A score of “3.5” is between “Somewhat not confident/Somewhat unlikely” and “Neutral” indicating preference for a negative consciousness (e.g., low self-efficacy). Barriers for cancer surveillance, i.e., lack of physician referral, cost-related lack of access, fear of finding cancer, and perception that mammograms are unnecessary were also scored from 1 to 7 and a cut-off score of ≤5.5 was used to identify participants who reported low versus high barriers. A score of “5.5” is between “Neutral” and “Agree” and indicates the presence of a barrier. Personalized probabilities of developing cancer i.e., Gail and Claus scores were presented to relatives based on information provided in their baseline survey. Messages and tailored letters were reviewed for appropriateness and accuracy.

## 2. Results

Recruitment, enrollment, randomization and retention are shown in a consort diagram (Figure 1). Response rates were 38.6% for White/Other and 27.5% for Black, and 801 YBCS were allocated to study arms. YBCS identified 1875 eligible relatives and they were willing to contact 1360 (72.5%). The study invited *n* = 853 relatives (up to two relatives per YBCS); *n* = 442 (51.5%) accepted participation and *n* = 431 relatives were allocated to study arms. Overall, 11.9% YBCS and 27.4% relatives resided in 23 and 27 different U.S. states, respectively.

After randomization, YBCS and relatives in both study arms did not differ at baseline (Table 2). YBCS were on average 51.11 (±5.74) years old and were diagnosed on average 40.13 (± 4.66) years old; approximately one in five had more than one cancer diagnoses [31]. Relatives were on average 43.35 (±11.93) years old. About one in five YBCS and one in five relatives reported cost-related lack of access to healthcare. Follow-up surveys were received from 610 YBCS (76.2% retention) and 352 relatives (81.7% retention).

### 2.1. (Cascade) Genetic Testing

Genetic testing was reported by 23% of YBCS and 3% of relatives at baseline, while 8 months later approximately 28% YBCS and 5% relatives reported having genetic testing at follow-up (Table 3). There were 40 new YBCS reporting having genetic testing and 9 new relative cases reporting cascade genetic testing between the baseline and the follow-up survey. From logistic regression, relatives in the tailored arm were more likely to report cascade genetic testing, although the difference was not statistically significant. From multiple linear regression analyses, YBCS in the tailored arm were more likely to report higher self-efficacy for genetic services (Beta = 0.480; CI: (0.026–0.933); *p* = 0.0205). Participants needing less support from providers were consistently more likely to report significant changes in outcomes related to use of genetic services, i.e., higher self-efficacy and higher intention for YBCS (Beta = 0.355; CI: (0.141–0.569); *p* = 0.002 and Beta = 0.490; CI: (0.334–0.645); *p* < 0.001), and higher self-efficacy for relatives (Beta = 0.375; CI: [0.063–0.686]; *p* = 0.002). YBCS who were older (Beta = 0.074; CI: (0.037–0.111); *p* < 0.001), Black (Beta = 0.984; CI: (0.747–1.221) *p* < 0.001), with cost-related barriers (Beta = 0.048; CI: (0.630–1.466); *p* < 0.001), and living further from genetic services (Beta = 0.014; CI: (0.007–0.022); *p* < 0.001) reported higher intention for genetic testing. (Appendix B
Table A1). 

### 2.2. Breast Cancer Surveillance/Screening

At baseline, 84.27% (Tailored 85.92%; Targeted 82.63%) of YBCS and 75.41% of relatives (Tailored 74.89%; Targeted 76.04%) reported CBE consistent with NCCN guidelines (Table 3). In the 8-month follow-up survey, 89.51% of YBCS (Tailored 90.70%; Targeted 88.33%) and 84.69% of relatives (Tailored 85.36%; Targeted 83.85%) reported having CBE consistent with NCCN guidelines. At baseline, 87.41% (Tailored 87.64%; Targeted 87.16%) of YBCS and 70.50% of relatives (Tailored 69.87%; Targeted 71.31%) reported having mammograms consistent with NCCN guidelines. In the 8-month follow-up survey, 91.41% of YBCS (Tailored 92.65%; Targeted 90.15%) and 79.86% of relatives (Tailored 80.77%; Targeted 78.69%) reported having mammograms consistent with NCCN guidelines. There was about 5% and 4% increase from baseline to follow up in CBE and mammography for YBCS, while the increase was close to 10% for both outcomes for relatives. Although there were not significant differences between the two arms, there were significant changes in CBE and mammography for YBCS and relatives within each intervention arm compared to baseline. From logistic regression analyses, YBCS needing more support from providers were less likely to report CBE compared with YBCS who needed less support (*OR* = 0.974; *CI*: (0.959–0.988); *p* < 0.002). Older relatives were more likely to report a mammogram compared with younger relatives (*OR* = 1.004; *CI*: (1.002–1.007); *p* < 0.002). From multiple linear regression analyses, YBCS without health insurance reported significantly higher self-efficacy for CBE and self-efficacy for *mammography* (*Beta* = 0.696; *CI*: (0.278–1.113); *p* < 0.001; *Beta* = 0.830; *CI*: (0.406–1.254); *p* < 0.001). Intention to have a mammogram increased for YBCS with a routine source of care (*Beta* = 1.052; *CI*: (0.784–1.320); *p* < 0.001) (Appendix B
Table A1).

### 2.3. Effects for Black and White/Other Participants

As shown in Table 4, there were not significant differences in genetic testing and surveillance/screening from baseline to follow-up between Black and White/Other participants (no differences between groups). Changes from baseline to follow-up were significantly different for both groups (significant within group differences).

### 2.4. Satisfaction with the Interventions

As shown in Table 5, approximately 66% of participants reported reading the intervention materials at least once. Separate intervention effects were examined for participants reporting not reading the intervention materials (*n* = 131; 74 YBCS and 57 Relatives) and the main findings remained consistent. Two out of three participants reported discussing intervention materials primarily with first-degree and with non-biological relatives, most often females and/or from the maternal side of the family. We compared acceptability and perceived usefulness for YBCS versus relatives; tailored versus targeted arm; and Black versus White/Other participants. Black participants reported significantly higher satisfaction, acceptability, and usefulness of the interventions, and getting information that helped them discuss ways to lower their breast cancer risk with their provider. Relatives requested significantly more information for breast cancer risk factors and screening.

## 3. Discussion

Uptake of genetic testing in both arms of our RCT increased approximately 5%, which is similar to a previous RCT reporting the efficacy of a booklet on rates of genetic testing [28]. This change of 5% is commendable, given that participants received intervention materials only once and had no contact with the healthcare system, in contrast to more resource intensive studies. Given that YBCS were on average 11 years post-diagnosis, it is unlikely that this change was due to the passage of time, but most likely can be attributed to exposure to the intervention materials. At the same time, YBCS in the tailored arm were more likely to report higher self-efficacy for genetic testing, more so than the targeted intervention. Thus, the tailored intervention generated added value, since self-efficacy is an important predictor of subsequent behavior [29]. Tailored feedback improves the impact of the message on health behaviors [32,33] because it addresses personal characteristics and needs, and increases attention and information processing [34]. The lower uptake of genetic testing among YBCS may be related to other factors, including the short-term follow-up, the recruitment strategy precluding a referral from a healthcare provider, and the fact that YBCS were on average 11 years post diagnosis and genetic testing may not have been perceived as relevant or urgent [35]. Furthermore, relatives’ eligibility for cascade genetic testing depends first, on the YBCS having genetic testing as the affected relative, and second, on the YBCS’ test identifying a pathogenic variant associated with HBOC. Rates of cascade genetic testing among relatives might have been higher, if the study included the 58 YBCS reporting a known pathogenic variant in themselves or in another relative. Moreover, it is harder to improve rates of genetic testing for the 163 YBCS who had testing prior to the study, thus, influencing intervention outcomes for YBCS and relatives.

There was no difference in participant satisfaction between the two interventions. Since rates of genetic testing at baseline were low and there was little variation among study arms for this key outcome, it would be interesting to study if our targeted booklet yields better rates of genetic testing when integrated in the healthcare system and the message were reinforced by provider referrals. Future studies should also perform a cost-effectiveness analysis of tailored versus targeted interventions for genetic testing [35], since targeted messages may be as effective but less resource intensive than tailored interventions. Tailored efforts may need to focus on increasing participation of YBCS in similar initiatives and cascade genetic testing among relatives. A stepped approach with personal and timed follow-up contacts for those who do not respond to the initial invitation and those needing greater support from providers may prove efficacious and cost-effective.

An important finding of the RCT was that there was 5% to 10% increase from baseline to follow-up in CBE and mammography rates among YBCS and relatives in both study arms. Since they have already been diagnosed with the disease, YBCS are more likely to have an established relationship with the healthcare system and have cancer surveillance according to guidelines. In contrast, cancer-free relatives may not have a routine source of care and an established relationship with a provider who would guide their screening practices. Self-efficacy and intention for surveillance, which are important predictors of subsequent behavior [29], increased consistently for subgroups of participants, especially those who did not need support from providers. The booklet and the letters were an efficient and low-resource strategy for increasing screening. Given the minimal contact with participants, and that at baseline 80% of YBCS and 70% of relatives reported previous CBE and mammography leaving less room for improvement, the outcomes of the RCT indicate that both interventions addressed appropriate theory-based factors that help increase screening behaviors. Alternatively, the Healthy Michigan Plan (Medicaid expansion), which was enacted in April 2014, might have helped mitigate cost-related barriers and granted access to genetic testing and surveillance to uninsured individuals, most of whom belong to minority groups [23,36,37].

This study included a large sample of Black YBCS. Black YBCS were more likely to report higher self-efficacy and higher intention for genetic testing, higher satisfaction with their participation in the study and intervention materials, and needing additional information about genetic services and breast cancer screening compared to White/Other participants. Taken together these findings suggest that intervention booklets and letters achieved higher acceptability and perceived usefulness among Black participants, which can increase effectiveness in special populations [38,39,40]. Black participants in our study reported that underutilization of genetic services was due to lack of physician referrals and cost-related barriers [41,42,43]. Intervention materials partially addressed these barriers by encouraging participants to initiate a genetic evaluation, and by providing information about costs of genetic testing and access to low cost mammograms. Our booklets can empower minority communities and engage them in health policies for genetic screening [44].

Strengths of the study are the study design and the partnership between a state health department and a leading academic institution. Advantages of recruiting from a state cancer registry are the ability to identify retrospectively a large number of potentially eligible subjects, from diverse geographical areas and racial/ethnic backgrounds, enroll them in prospective trials, and produce results that are more representative. A disadvantage was the lower participation rate compared to recruitment from clinical sites [45,46]. However, a response rate of approximately 30% is common for RCTs recruiting participants from central cancer registries [45,46,47]. Comparisons between responders and non-responders was not possible due to lack of data for non-responders. Our RCT was underpowered to detect outcomes among Black relatives, despite their larger sample compared to previous studies. Since there was not a “no treatment” group, we can only conclude that there was no difference between the two interventions. However, the increase in genetic testing and surveillance was similar to other interventions that were compared to a “no treatment” group. It is possible that eight months was not adequate time to observe changes in outcomes. Limitations were a possible recall bias and that participants could not be blinded to study allocation, although they were not aware of materials delivered in the other arm. Additional limitations include possible recall bias, not assessing if participants received counseling but declined testing, and that information about genetic services might not be relevant for the 11.9% YBCS and 27.4% relatives not living in Michigan. Finally, findings cannot be generalized to men, to women older than 64 years old, and women who are pregnant, imprisoned, institutionalized, or non-English speaking.

## 4. Materials and Methods

### 4.1. Design and Sample

This two-arm cluster RCT was conducted in the state of Michigan (NCT01612338); the protocol and study methodology have been previously published [48]. Institutional Review Boards of the University of Michigan (HUM00055949) and the Michigan Department of Health and Human Services (201202-09-EA) approved the study protocol. The Michigan Cancer Surveillance Program identified approximately 9000 women diagnosed with breast cancer between 20 and 45 years old from the cancer registry, who were eligible for genetic evaluation due to young age of cancer onset. Cases of men with breast cancer identified in the cancer registry were not included due to their small number. Black YBCS were separated to form a separate stratum. Approximately 7% of YBCS of other racial/ethnic backgrounds (e.g., Arab Americans etc.) were grouped with White YBCS, because they could not form a separate stratum. A computer algorithm randomly selected a stratified sample of 3000 YBCS based on their cancer registry index number (1500 Black and 1500 White/Other) with oversampling of Black YBCS.

YBCS were eligible to participate if they were 20 to 45 years old when diagnosed with invasive breast cancer or ductal carcinoma in situ; 25 to 64 years old at the time of the study; Michigan residents at the time of diagnosis; and able to read English and provide informed consent. Female relatives had to be cancer-free and 25 to 64 years old; able to read English and provide informed consent; and YBCS would be willing to contact them. Up to two relatives per YBCS were included. Priority was given to younger and first-degree relatives [31]. YBCS and relatives had to be older than age 25 to assess their surveillance behavior according to NCCN guidelines. The upper age limit was set at 64 due to more limited insurance coverage for older individuals that may hinder surveillance. Excluded were pregnant, incarcerated, or institutionalized participants since they may not get mammograms.

### 4.2. Randomization and Masking

The Michigan Cancer Surveillance Program inquired with the reporting facility and physician of record whether there was any reason that an YBCS should not be contacted. If a response was not received within 30 days, a recruitment package was mailed to the YBCS. Eligible YBCS received up to three mailed invitations over a period of four months. YBCS who accepted participation were asked in the baseline survey if they were willing to invite their first- and second-degree female relatives to take part in the study. In order to alleviate ethical concerns in contacting relatives without their explicit consent, recruitment materials were mailed to YBCS, who passed them on to relatives. When YBCS reported they already had genetic testing, a certified genetic counselor contacted them by phone to double-check that their response was accurate. There was *n* = 58 YBCS who reported that they or one of their relatives had a pathogenic variant in *BRCA* or other gene associated with hereditary breast cancer, or had another hereditary cancer syndrome (e.g., Li Fraumeni). These YBCS were provided appropriate information and were excluded from the RCT because intervention materials were not applicable. There was *n* = 163 YBCS who reported a negative genetic test result. These YBCS and their relatives (*n* = 103) were included. None was a “true negative” and we could not exclude the possibility that there might be updated information to justify a new genetic evaluation. We could also not exclude the possibility of a pathogenic variant in relatives.

YBCS and relatives had to return a signed informed consent before receiving the baseline survey. Recruitment of YBCS and relatives took place over six months from the date of mailing the first invitation letter to reduce bias due to sample “maturation”. Research staff at the cancer registry used a computer-generated algorithm to randomize (1:1) YBCS and relatives as stratified (Black vs. White/Other) family units (i.e., dyads and triads) and allocate them to one of the two study arms. Research staff at the cancer registry were not involved in data analyses examining efficacy of the two interventions. YBCS and relatives were randomized as stratified (Black vs. White/Other) family units (i.e., dyads and triads) to one of the two study arms (1:1) using a computer-generated allocation algorithm. All members of a family unit received intervention materials at the same time by postal mail and participants were unaware of the intervention materials delivered to the other study arm. Participants received $10 gift cards for completing the baseline survey and $20 gift cards for the follow-up survey, respectively. The study employed two research staff at 40% Full Time Equivalent (FTE) for six months for recruitment, mailing, and generating intervention materials. A certified genetic counsellor (10% FTE for three months) conducted risk assessments, and verified YBCS’ self-reports of being a mutation carrier and the content of intervention materials.

### 4.3. Data Collection and Measures

Eligible YBCS were mailed a baseline survey (Time 1). Following assessment of their baseline information, their relatives were recruited, and family units were randomized to the targeted or tailored intervention. The follow-up survey (Time 2) was mailed to participants approximately 8 months after the intervention to allow sufficient time for pursuing the primary outcomes within the timeframe of the study. Research staff made two attempts via phone, mail, or email to contact YBCS and relatives if they did not return the follow-up survey within six weeks.

Table 6 describes the instruments used to assess genetic testing and breast cancer surveillance/screening. The research team determined consistency of surveillance with NCCN guidelines based on items from the Centers for Disease Control and Prevention: Behavioral Risk Factor Surveillance System: 2001 Survey Questions [49]. Two single items asked participants “how often you need advice from relatives/healthcare providers to engage in behaviors aiming to find cancer at an early stage” (Likert scale 1 = never to 7 = always). The 8-month follow-up survey included additional questions assessing whether the interventions provided new and helpful information, and examined intervention acceptability, interest, usefulness, level of detail, relevance, and satisfaction (Likert scale 1 = low to 7 = high) [50,51].

### 4.4. Sample Size and Power Evaluation

Using data from previous mammography RCTs [23,25], we calculated a sampling size that is expected to ensure 80% power to detect a small (Cohen’s d = 0.2) to medium (Cohen’s d = 0.5) effect, i.e., difference in intervention effect size between group means (d = 0.3) or between percentages (h = 0.3), using a two-tailed test with a false positive rate of α = 0.05 [60]. Power analysis with PASS software [61] determined that after attrition 176 participants were needed per group or 352 in total.

### 4.5. Statistical Analyses

Statistical analyses were performed using R 3.4.4. (R Core Team. R: A language and environment for statistical computing. R Foundation for Statistical Computing, Vienna, Austria, available from https://www.r-project.org/). Descriptive analyses compared means and proportions in demographic and clinical factors between and within intervention groups across time (baseline and 8-month follow-up). Differences between intervention arms were tested at baseline and follow-up with two proportions z-test for proportions and with t-test for means. We performed separate analyses for YBCS and relatives for genetic testing, CBE, and mammography. 

We conducted multiple linear regressions to explored associations between outcomes (self-efficacy and intention for genetic testing, CBE, and mammography) and predictor variables for YBCS and relatives including intervention grouping, antecedents, and barriers. Changes in frequencies for outcomes were demonstrated after using Intention-To-Treat (ITT), defined as “once randomized, always analyzed” [62]. Similar to last observation carried forward (LOCF), ITT is a commonly used approach in RCTs that addresses noncompliance and missing outcomes. ITT can reduce the impact from lost to follow-up but also dilutes intervention effects, making a generally conservative estimate. Outcomes reported from drop out cases in the baseline survey were carried forward to the follow-up survey. This approach avoids overoptimistic estimates resulting from removing dropout cases. Comparisons between interventions or racial groups for genetic testing and surveillance were conducted with two-proportion z-test. Fisher’s Exact test was used for small samples. McNemar’s test was used for comparisons within interventions or racial groups, and McNemar’s Exact test for small samples. Confidence intervals were computed for parameter estimates [63]. Acceptability and perceived usefulness of the interventions for YBCS versus relatives, for each intervention arm, and for Black versus White/other participants were tested using parametric t-tests, and *p*-values were adjusted for multiple testing via Bonferroni corrections. Two sensitivity analyses were performed for within group comparisons: (1) excluding whoever had genetic testing at baseline but keeping dropouts (ITT); (2) excluding both whoever had genetic testing and whoever dropped out. Both analyses had shown similar results as presented in Table 3. Keeping a large baseline population (true number of subjects who received the interventions) and ITT are both conservative approaches.

We examined core features of missing data (<18% of multi-item scales) and cases who dropped out (*n* = 270, 21.92%) from the follow-up survey. No special patterns of missing values were identified. We used demographic variables to examine if there was a clear pattern of lost-to-follow-up across subgroups using machine-learning approaches [64,65,66]. The results indicated random drop out patterns across subgroups, thus, multiple imputations addressed missing values for subsequent analyses. Two imputation approaches were used; LOCF and multiple imputations for different analyses steps. LOCF was conducted for between group and within group comparisons for dichotomous outcomes (genetic testing, CBE and mammography), while multiple imputations were conducted for multiple linear regression modeling associations between intervention effects for continuous variables (self-efficacy and intention for genetic testing, surveillance and predictor variables for YBCS and relatives including intervention grouping, antecedents, and barriers). All variables in the final model were used to generate three imputed datasets. *p*-values were pooled across the three models built using the three imputed datasets and adjusted for multiple testing using Bonferroni corrections.

## 5. Conclusions

This RCT is aligned with evidence-based recommendations for public health action relevant to cancer predisposition cascade genetic screening [67,68]. Adoption of these recommendations will achieve a population-level reduction in cancer morbidity and mortality. A combination of targeting and tailoring health messages and recruitment efforts will likely maximize resources [69] and help achieve optimal outcomes for genetic testing and cancer surveillance.

## Figures and Tables

**Figure 1 cancers-12-02526-f001:**
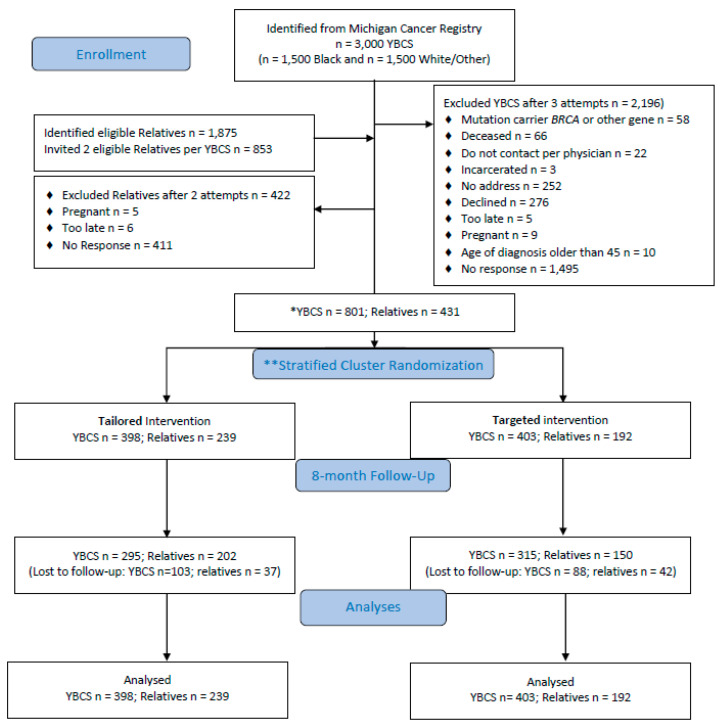
Consort diagram. * YBCS = young breast cancer survivor; ** Stratified randomization of YBCS according to race (Black vs. White/Other); relatives follow randomized arm of YBCS.

**Table 1 cancers-12-02526-t001:** Elements of the tailored and the targeted interventions.

Adapted TPB *	Tailored Intervention		Targeted Intervention	
	**Booklet 1—Surveillance and Genetic Testing ****			
Knowledge	Risk factors and cancer genetics		Risk factors and cancer genetics	
	Breast cancer surveillance		Breast cancer surveillance	
Self-efficacy screening and genetic services	Genetic counseling, cost		Genetic counseling, cost	
	CBE and Mammography, sources for low cost screening		CBE and Mammography, sources for low cost screening	
	Certified genetic services in MI		Certified genetic services in MI	
	**Booklet 2—Family Support ****			
Subjective norms	Cancer and open family communication			
	Family support in illness			
	**Tailored Letter**		**Targeted Letter**	
	**YBCS**	**Relatives**	**YBCS ****	**Relatives**
Knowledge	Surveillance according to guidelines for follow-up care	Screening according to guidelines for breast cancer	NCCN guidelines for follow-up care	NCCN guidelines for screening
Attitudes	Barriers/facilitators to follow-up care	Barriers/facilitators to screening	Increased risk - early age of cancer onset	Increased risk - family history
	Barriers/facilitators to genetic services	Barriers/facilitators to genetic services	Suggest genetic evaluation	Suggest genetic evaluation
	Fear of cancer recurrence	Gail and Claus risk scores		
	Genetic literacy, breast cancer risk factors, inheritance	Genetic literacy, breast cancer risk factors, inheritance		
Subjective norms	Family communication	Family communication		
	Family support in illness	Family support in illness		

* TPB = Theory of Planned Behavior, ** Provided as Appendix A.

**Table 2 cancers-12-02526-t002:** Demographic characteristics and barriers by intervention arm (%) or mean ± SD.

YBCS *	Demographics	Baseline *n* = 801		Follow-Up *n* = 610	
		Tailored *n* = 398	Targeted *n* = 403	Tailored *n* = 295	Targeted *n* = 315
Antecedents	Age (range 25–64)	51.58 ± 5.73	50.65 ± 5.76	51.76 ± 5.64	51.17 ± 5.51
	Race (Black %)	162 (40.70%)	162 (40.20%)	98 (33.22%)	116 (36.83%)
	Education ≤ High School	85 (21.36%)	103 (25.56%)	65 (22.03%)	78 (24.76%)
	Caregiving responsibilities	120 (30.15%)	141 (34.99%)	71 (24.07%)	89 (28.25%)
	Anxiety	102 (25.63%)	122 (30.27%)	80 (27.12%)	94 (29.84%)
	Depression	109 (27.39%)	116 (28.78%)	91 (30.85%)	91 (28.89%)
	Comorbidities	252 (63.32%)	277 (68.73%)	190 (64.41%)	211 (66.98%)
Barriers **	Income ≤ $40,000	118 (29.65%)	124 (30.77%)	90 (30.51%)	95 (30.16%)
	No insurance	30 (7.54%)	22 (5.46%)	15 (5.08%)	17 (5.40%)
	No routine source of care	23 (5.78%)	33 (8.19%)	20 (6.78%)	16 (5.08%)
	Cost-related lack of access	73 (18.34%)	71 (17.62%)	42 (14.24%)	43 (13.65%)
	Mean distance to closest genetic center (miles)	18.58 ± 26.48 (0–147.6)	19.51 ± 27.38 (0–147.6)	18.58 ± 26.45 (0–147.6)	19.24 ± 27.10 (0–147.6)
**RELATIVES**	**Demographics**	**Baseline *n* = 431**		**Follow-Up *n* = 352**	
		**Tailored** ***n* = 239**	**Targeted** ***n* = 192**	**Tailored** ***n* = 202**	**Targeted** ***n* = 150**
Antecedents	Age (range 25–64)	43.64 ± 12.05	43.00 ± 11.69	43.45 ± 12.14	43.23 ± 11.86
	Race (Black %)	46 (19.25%)	41 (21.35%)	33 (16.34%)	32 (21.33%)
	Education ≤ High School	40 (16.74%)	32 (16.67%)	33 (16.34%)	27 (18.00%)
	Caregiving responsibilities	105 (43.93%)	80 (41.67%)	87 (43.07%)	58 (38.67%)
	Anxiety	72 (30.13%)	43 (22.40%)	55 (27.22%)	34 (22.67%)
	Depression	62 (25.94%)	49 (25.52%)	54 (26.73%)	42 (28.00%)
	Comorbidities	138 (57.74%)	92 (47.92%)	115 (56.93%)	76 (50.67%)
Barriers **	Income ≤ $40,000	65 (27.20%)	70 (36.46%)	63 (31.19%)	55 (36.67%)
	No insurance	33 (13.81%)	23 (11.98%)	16 (7.92%)	16 (10.67%)
	No routine source of care	30 (12.55%)	16 (8.33%)	20 (9.90%)	9 (6.00%)
	Cost-related lack of access	52 (21.76%)	30 (15.63%)	42 (20.79%)	28 (18.67%)
	Mean distance to closest genetic center (miles)	21.16 ± 31.09 (0–196.7)	25.44 ± 33.41 (0–195.9)	21.16 ± 31.09 (0–196.7)	25.69 ± 33.65 (0–195.9)

* YBCS = young breast cancer survivor; ** Proportion of YBCS and Relatives who reported each barrier.

**Table 3 cancers-12-02526-t003:** Participants’ genetic testing, CBE, and mammography by intervention arms.

Outcomes for YBCS * Tailored *n* = 398 Targeted *n* = 403	Baseline		Follow-Up **		Tailored vs. Targeted *p* Value ^A^ (95% CI)	Change from Baseline to Follow-Up *p* Value ^B^ (95% CI)	
	Tailored	Targeted	Tailored	Targeted		Tailored	Targeted
Had Genetic Testing	79 (19.85%)	107 (26.55%)	99 (24.87%)	127 (31.52%)	1.00 (−0.030–0.031)	≤**0.001** ^b^ (0.031–0.077)	<**0.001** ^b^ (0.031–0.076)
CBE according to NCCN *** Guidelines	342 (85.92%)	333 (82.63%)	361 (90.70%)	356 (88.33%)	0.66 (−0.040–0.023)	<**0.001** ^b^ (0.029–0.074)	<**0.001** ^b^ (0.037–0.084)
Mammography according to NCCN *** Guidelines^1^	298 (87.64%)	292 (87.16%)	315 (92.65%)	302 (90.15%)	0.17 (−0.009–0.055)	<**0.001** ^b^ (0.029–0.079)	**0.002**^b^ (0.014–0.054)
**Outcomes for Relatives** **Tailored *n* = 239** **Targeted *n* = 192**	**Baseline**		**Follow-Up ****		**Tailored vs. Targeted** ***p* Value ^A^** **(95% CI)**	**Change from** **Baseline to Follow-Up** ***p* Value ^B^** **(95% CI)**	
	**Tailored**	**Targeted**	**Tailored**	**Targeted**		**Tailored**	**Targeted**
Had Genetic Testing	9 (0.04%)	4 (0.02%)	17 (0.07%)	5 (0.03%)	0.08 ^a^ (−0.001–0.058)	**0.008**^b^ (0.015–0.065)	**1**^b^ (0.000–0.029)
CBE according to NCCN *** Guidelines	179 (74.89%)	146 (76.04%)	204 (85.36%)	161 (83.85%)	0.44 (−0.032–0.085)	**<0.001** (0.069–0.151)	**<0.001**^b^ (0.044–0.125)
Mammography according to NCCN *** Guidelines ^2^	109 (69.87%)	87 (71.31%)	126 (80.77%)	96 (78.69%)	0.43 (−0.039–0.110)	**<0.001**^b^ (0.065–0.168)	**0.004**^b^ (0.034–0.135)

* YBCS = young breast cancer survivor; ** Intention to Treat; *** NCCN = National Comprehensive Cancer Network; ^A^ Two-proportions z-Test or ^a^ Fisher’s Exact Test; ^B^ McNemar’s test or ^b^ McNemar’s Exact Test; ^1^ Tailored *n* = 340 and Targeted *n* = 335 after excluding YBCS with double mastectomy who do not receive mammograms per NCCN guidelines (excluded Tailored *n* = 58; Targeted *n* = 68); ^2^ Tailored *n* = 156 and Targeted *n* = 122 after excluding relatives younger than 35 years old AND relatives between 35 and 40 with Gail lifetime risk <20% who do not receive mammograms per NCCN guidelines (excluded Tailored *n* = 83; Targeted *n* = 70). Bold = statistical significance.

**Table 4 cancers-12-02526-t004:** Participants’ genetic testing, CBE, and mammography by race.

Outcomes for YBCS * Black *n* = 324 White/Other *n* = 447	Baseline		Follow-Up **		Black vs. White/Other *p* Value ^A^ (95% CI)	Change from Baseline to Follow-Up *p* Value ^B^ (95% CI)	
	Black	White/Other	Black	White/Other		Black	White/Other
Had Genetic Testing	52 (16.05%)	134 (28.09%)	68 (20.99%)	158 (33.12%)	0.92 (−0038–0.054)	<**0.001** ^b^ (0.028–0.079)	<**0.001** ^b^ (0.035–0.079)
CBE according to NCCN *** Guidelines	268 (82.72%)	407 (85.32%)	286 (88.27%)	431 (90.36%)	1 (−0.033–0.036)	<**0.001** ^b^ (0.033–0.086)	<**0.001** ^b^ (0.035–0.079)
Mammography according to NCCN *** Guidelines ^1^	244 (83.28%)	346 (90.58%)	259 (88.40%)	360 (94.24%)	0.46 (−0.020–0.049)	<**0.001** ^b^ (0.029–0.083)	<**0.001** ^b^ (0.020–0.061)
**Outcomes for Relatives** **Black *n* = 87** **White/Other *n* = 344**	**Baseline**		**Follow-Up ****		**Black vs. White/Other** ***p* Value ^A^** **(95% CI)**	**Change from** **Baseline to Follow-Up** ***p* Value ^B^** **(95% CI)**	
	**Black**	**White/Other**	**Black**	**White/Other**		**Black**	**White/Other**
Had Genetic Testing	2 (2.30%)	11 (3.20%)	4 (4.60%)	18 (5.23%)	1.00 ^a^ (−0.035–0.039)	**0.5**^b^ (0.003–0.081)	**0.016**^b^ (0.008–0.041)
CBE according to NCCN *** Guidelines	63 (72.41%)	262 (76.16%)	71 (81.61%)	294 (85.47%)	1.00 (−0.076–0.068)	**0.008**^b^ (0.041–0.173)	**<0.001** (0.064–0.129)
Mammography according to NCCN *** Guidelines ^2^	39 (65.00%)	157 (72.02%)	45 (75.00%)	177 (81.19%)	1.00 (−0.085–0.102)	**0.031**^b^ (0.038–0.205	**<0.001**^b^ (0.057–0.138)

* YBCS = young breast cancer survivor; ** Intention to Treat; *** NCCN = National Comprehensive Cancer Network; ^A^ Two-proportions z-Test or ^a^ Fisher’s Exact Test; ^B^ McNemar’s test or ^b^ McNemar’s Exact Test; ^1^ Tailored *n* = 293; Targeted *n* = 382, after excluding YBCS with double mastectomy (excluded Tailored *n* = 31; Targeted *n* = 95); ^2^ Tailored *n* = 60; Targeted *n* = 218, after excluding relatives younger than 35 years old AND relatives between 35 and 40 with Gail lifetime risk <20% according to NCCN guidelines (excluded Tailored *n* = 27; Targeted *n* = 126). Bold = statistical significance.

**Table 5 cancers-12-02526-t005:** Evaluation of the acceptability and perceived usefulness of the interventions for YBCS vs. Relative; for Tailored vs. Targeted; and for Black vs. White/Other.

I Discussed the Information in the Booklet(s) and Letter with… (Multiple Choice)					Count		
No one					324		
Not a biological relative (spouse, in laws, friend)					323		
First degree relatives (mother, father, sister, brother, children)					700		
Second degree relative (grandmother, grandfather, grandchildren, aunts, uncles, nephews, nieces)					163		
First cousins					65		
Healthcare provider (oncologist, genetic specialist, nurse, primary care provider)					124		
Other					5		
**The Brochures and Letter I Received…** **(1–7) (Mean Score)**	**Overall**	**YBCS ****	**Relatives**	**Tailored**	**Targeted**	**Black**	**White/Other**
…provided me with new information	4.84	4.77	4.94	4.81	4.87	**5.07**	**4.74**
…provided helpful information	5.15	5.16	5.14	5.14	5.17	**5.36**	**5.07**
…were overall easy to understand, important, useful, and interesting *	5.04	5.05	5.04	5.06	5.02	**5.35**	**4.93**
…helped me talk with my healthcare provider about my breast cancer risk	4.26	4.24	4.32	4.28	4.25	**4.74**	**4.07**
…helped me talk with my provider about ways to lower my cancer risk	4.23	4.21	4.25	4.22	4.23	**4.70**	**4.02**
**I Would Like to Get More Information about… (1–7) (Mean score)**	**Overall**	**YBCS ****	**Relatives**	**Tailored**	**Targeted**	**Black**	**White/Other**
…risk factors for breast cancer	4.87	**4.67**	**5.22**	4.87	4.88	**5.39**	**4.66**
…importance of family history for cancer risk	4.90	**4.71**	**5.22**	4.83	4.98	**5.46**	**4.67**
…genetic counseling and genetic testing	4.83	4.73	5.02	4.75	4.92	**5.47**	**4.57**
…where to get genetic counseling and testing	4.70	4.58	4.90	4.67	4.74	**5.39**	**4.41**
…breast cancer screening	4.86	**4.71**	**5.10**	4.86	4.86	**5.43**	**4.63**
…low cost breast cancer screening	4.52	4.37	4.75	4.37	4.68	**5.29**	**4.20**
…family communication in breast cancer	4.26	4.18	4.41	4.13	4.41	**5.04**	**3.95**
…family support in breast cancer	4.22	4.14	4.36	4.11	4.34	**4.98**	**3.91**
I would suggest the study to other women like me	5.77	5.81	5.70	5.77	5.77	**6.05**	**5.66**
The study was important	6.16	6.16	6.16	6.22	6.10	**6.37**	**6.08**
I benefited from taking part in the study	5.57	5.51	5.67	5.61	5.53	**5.97**	**5.40**

* average of 16 items; ** YBCS = young breast cancer survivor; Bold = significant difference from *t*-test with Bonferroni corrections.

**Table 6 cancers-12-02526-t006:** Measures used to assess covariates and outcomes.

	Instrument	YBCS		Relative	
		Baseline	Follow-Up	Baseline	Follow-Up
**Demographics**					
Age, Race, Education	Behavioral risk factors surveillance system [49]	√		√	
Income, Insurance	Behavioral risk factors surveillance system [49]	√		√	
Routine source of care	Coordination of medical care (multiple choices)	√		√	
Cost-related lack of access	High out-of-pocket costs (yes/no)	√	√	√	√
Distance—genetic services	Great Circle Distance Formula [52]	√		√	
Caregiving responsibilities	Lives with children under 18 years old and/or with elderly parents	√		√	
**Health history**					
Anxiety, Depression, Comorbidities	Anxiety, Depression, and 11 chronic conditions associated with mobility (yes/no) [53]	√	√	√	√
Cancer and family history	Behavioral risk factors surveillance system (validated) [49]	√	√	√	√
Surgery	American Society of Clinical Oncology (ASCO) guidelines [4]	√	√		
Reproductive history	Risk factors associated w/the Gail and the Claus models [54,55,56]			√	√
**Family characteristics**					
Family coherence	Family Hardiness Index, 20 items, 7-point Likert scale [57]	√		√	
**Facilitators and barriers**					
Barriers for mammography	Decisional balance scale for mammography, 20 items, 7-point Likert scale [58]	√	√	√	√
Perceived expectations of healthcare providers/family members	1 item, 7-point Likert scale *“Do you believe that your healthcare providers/relatives want you to get (genetic testing) to find cancer at an early stage?”*	√		√	
Motivation to comply with recommendations from healthcare providers/family members	1 item, 7-point Likert scale *“How often do you try to do what your healthcare providers/relatives want you to do about finding cancer at an early stage?”*	√		√	
**Genetic services and breast cancer surveillance**					
Genetic services (testing)	NCCN Guidelines [59]	√	√	√	√
Cancer surveillance (CBE, mammography)	NCCN Guidelines [59]	√	√	√	√
Self-efficacy (genetic testing, CBE, mammography)	1 item, 7-point Likert scale *“During the next 12 months how confident do you feel in your ability to ask your healthcare provider for (genetic testing/CBE, mammography).”*	√	√	√	√
Intention (genetic testing, CBE, mammography)	1 item, 7-point Likert scale *“During the next 12 months how likely are you to ask your healthcare provider if (genetic testing/CBE/mammography) is a right test for you.”*	√	√	√	√

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
