# Peer review of "Genetic Testing and Surveillance of Young Breast Cancer Survivors and Blood Relatives: A Cluster Randomized Trial"

_cancers, 2020, doi:10.3390/cancers12092526_

Round 1

Reviewer 1 Report

This study aimed to compare a tailored and a targeted intervention that designed to increase genetic testing, clinical breast exam (CBE), and mammography in young breast cancer survivors (YBCS) and their blood relatives. This cluster randomized study recruited subjects from a cancer registry and enrolled them in the trial. I totally agree that this research issue is interesting. However, from my own experience, the methodology and relevant findings were not clear addressed in the manuscript. Therefore, I recommend authors to address clear methodology in the manuscript and provide more detail results. Followings were some major and minor concerns.

Major Concerns:

-- The present RCT compared the efficacy of two low-resource interventions delivered via postal mail, which included a targeted (more generic) versus a tailored (person-specific) intervention. The outcomes presented in this paper are initiation of genetic testing for YBCS, cascade genetic testing for relatives, and surveillance (CBE and mammography screening) consistent with national guidelines for YBCS and relatives. Therefore, I considered that authors attempted to assure whether tailored intervention was much more efficient than the general intervention. According to present findings (Table 3), the distributions of genetic testing, CBE, and mammography did not show the statistically significant between tailored group and targeted group. However, authors still stated some statements in Results such as follows: “YBCS in the tailored arm reported higher self-efficacy for genetic services (p=0.0205)” The statements in Results section and the findings in Tables were inconsistent. It was very confusing to the readers. I considered that some data were not shown in the manuscript. I strongly recommend authors to revise their Tables and highlight their main findings.

--This two-arm cluster RCT was conducted in the state of Michigan. In this study, YBCS were eligible to participate if they were 20 to 45 years old when diagnosed with invasive breast cancer or ductal carcinoma in situ; 25 to 64 years old at the time of the study; female relatives had to be cancer-free and 25 to 64 years old; able to read English and provide informed consent; and YBCS would be willing to contact them. However, it was still unclear why authors included the ‘female relatives’ in this study. Authors only carried some analyses in YBCS population and female relative population, separately. If authors did not undergo further analyses, why not pool all the participants into one population? Author should provide clear explanation.

-- Authors have emphasized the relevant findings in Abstract: “At 8-months follow-up YBCS in the tailored arm reported higher self-efficacy for genetic services (p=0.0205). Genetic testing increased approximately 5% for YBCS in the tailored and the targeted arm (p≤0.001; p<0.001) and for Black and White/Other YBCS (p<0.001; p<0.001). CBEs and mammograms increased significantly in both arms, 5% for YBCS and 10% for relatives and were similar for Blacks and White/Others”. However, authors did not display the relevant Tables (regression models considering the covariates?) in Results Section. Authors only addressed the statements as follows: “Participants needing less support from providers were consistently more likely to report significant changes, i.e., higher self-efficacy and higher intention for YBCS (Beta=0.355; CI:[0.141-0.569]; p=0.002 and Beta=0.490; CI:[0.334-0.645]; p<0.001), and higher self-efficacy for relatives (Beta=0.375; CI:[0.063-127 0.686]; p=0.002)…” I recommend authors to provide clear Tables for all relevant analyses (regression models considering the covariates) in order to avoid the misleading.

-- Intention to treat methodology was used throughout the manuscript. Authors should address the relevant statements and provide some information in the Method section.

--In the Method section (Randomization and Masking of Materials), authors addressed relevant statements as follows: “YBCS who accepted participation were asked in the baseline survey if they were willing to invite their first- and second-degree female relatives to take part in the study.” We can imagine that young woman with breast cancer who be more or less likely to participate. Self-selection bias might occur when study participants could decide themselves to take part in or to reject to participate a study. Additionally, “A computer algorithm randomly selected a stratified sample of 3,000 YBCS...” However, the final sample size was only 801 YBCS who were allocated to study arms. Base on the above two reasons, the efficacy of the outcome might be less representative. Therefore, I recommend authors to address this limitation in Discussion.

--Authors addressed relevant statements as follows: “A certified genetic counsellor (10% FTE for three months) conducted risk assessments, and verified YBCS’ self-reports of being a mutation carrier and the content of intervention materials.” and “Research staff made two attempts via phone, mail, or email to contact YBCS and relatives if they did not return the follow-up survey within six weeks.”

I am wondering if the research staff and a certified genetic counsellor were unblinded and still concerned that the research staff might influence study participants to answer when the follow-up procedures for detecting the outcomes, such as self-efficacy and intention for genetic testing and surveillance. Did detection bias exist? Please addressed the relevant issue in the manuscript.

--Author concluded that “a combination of targeting and tailoring health messages and recruitment efforts will likely maximize resources and help achieve optimal outcomes for genetic testing and cancer surveillance”. The present findings could not support the Conclusion.

Minor Concerns:

--Table 1. should be use footnotes to define abbreviations and acronyms or display other explanatory notes (i.e. TPB).  

--The Bold form of Table 2. needs to be explain the representative meaning.

--In the Randomization and Masking of Materials of Method, “The study employed two research staff at 40% FTE for six months for recruitment, mailing, and generating intervention materials.” Please kindly define the full meaning of this abbreviation FTE.

Author Response

Please see the attached document for point by point response to the comments of Reviewer 1.

Reviewer 2 Report

Thanks for the opportunity to review your manuscript.

I commend the authors on their effort in undertaking this important public health research project.

I have a few comments and suggestions to help improve your manuscript.

Minor:

Title: I wonder if your title should be written the other way round, for example, “Genetic testing and surveillance of young breast cancer survivors and blood relatives: A cluster randomized trial”

Keywords are fine.

Abstract: ok

Introduction: There was a mention of a lack of awareness and/or access to genetic testing, line 43. I would love to see this unpacked a bit more. Why is this the case, socioeconomic statuses, failure in communication or what?

Line 91- why and how did you choose the cut-off of ≤5.5 for perceptions.  Was this random or is there a logical reason why bearing in mind that you chose a midpoint cut off of ≤3.5 for Self-efficacy and intention for genetic testing and surveillance.

Methods

Overall, very well written methods.

Can you please state which statistical analytical software/software were used for your analysis.

Results

Generally very well presented, I like the amount of details in your tables and the footnotes (with details of tests and summary) as well.

Other Comment:

Did you try to analyse the data by grouping into black, white and other (which you stated were mostly Arab-Americans).

Discussion:

Line 244- I am surprised with the statement, “Our RCT was underpowered…” You did compute a sample size calculation as stated on page 13, lines 321 to 326. One would expect you would have adjusted for all possible confounders prior to choosing a minimum sample size to detect a difference. Following from that assumption, it is reasonable to state that there is actually no difference in outcome.

Best wishes.

Author Response

Please see the attached document for a point by point response to comments of Reviewer 2

Reviewer 3 Report

This manuscript describes a cluster randomized trial comparing two interventions designed to improve rates of genetic testing, clinical breast examination, and mammography in young breast cancer survivors and their relatives.  The study was undertaken in Michigan, and deliberately oversampled Black women. The study addresses an important issue, statistical analyses used were appropriate for the research question and the results and discussion were well-written. A small number of suggestions and comments follow:

 Abstract, page 1, lines 17 – 31: This abstract provides a clear and concise summary of the trial design, methods, results, and conclusions. Some further details regarding statistical methods would be an important addition, in particular, including that intention to treat analysis was undertaken – a strength of the study, and important with regards to results.

 Introduction, page 1 – 2, lines 36 – 66: The introduction was well-written. The scientific background is outlined with clarity and the rationale for the study is evident. It would have been useful to include some discussion regarding the global burden of breast cancer among women of the sampled age group and how this compares with other regions of the US or globally. Importantly, the specific objectives of this cluster randomized controlled trial were described, along with a clear indication of outcomes.

Interventions, page 2 – 3, lines 67 – 95: It may be of value to include the relevant booklets as Supplementary materials to enable replication, in addition to Table 1.

Results, page 3, lines 97 – 102:  The written description of participants was clear. Notably, the response rate was low, however this is noted and addressed in the Limitations section of the study.

Results, page 4, lines 107 – 108: Was age normally distributed? If so, the mean is a useful measure to report for age, however would urge authors to include the standard deviation too. If the age distribution is non-normal, it may be more appropriate to describe the age using a median and interquartile range.

 Results, page 6, lines 138 - 139:

  • The odds ratios reported are very close to i.e., the null value, and therefore I query whether these results are in fact indicative of a clinically significant difference.
  • If it is felt that the above odds ratios are indicative of clinically significant differences, I suggest editing for clarity, as follows: “From logistic regression, YBCS needling less support from providers were more likely to report CBE (OR=0.974; [95% CI 0.959 - 0.988] compared with …)
  • Similarly, for the following sentence, i.e., “Older relatives were more likely to report a mammogram (OR=1.004; [95% CI 1.002-1.007) compared with…
  • Could the authors provide clarity around the increase in likelihood to report CBE, among YBCS needling less support – at three decimal points, the OR is below 1, suggesting a decrease in the likelihood to report CBE, rather than an increase?

 Results, page 6, line 146, table 3: Table 3 is difficult to follow with reference to the text at 2.2, Breast cancer surveillance / screening. This may be a formatting issue, however for clarity, it would be helpful to update the text at line 132 with the exact %, rather than approximations. This will aid clarity and flow of reading.

Results, page 8, lines 167 – 173: The results reported here are particularly interesting. The paper states that separate intervention effects were examined for participants reporting reading and not reading the intervention materials, and that the main results remained the same. Could the authors clarify if those who reported not reading the material discussed the intervention materials with others who had, but did not read them? If so, does this suggest that acquiring the relevant information is what is important, but that for some the mode of information acquisition is important? I.e., for some, verbal, personal interaction to acquire information is more accessible than reading information independently?

 Discussion, page 10 - 11, lines 188 – 251: Overall, the interpretation provided is consistent with results. Important points were made, including: that the 5% change was unlikely due to the passage of time (186 – 191); that it would be interesting to study if the targeted booklet yielded better rates of genetic testing when integrated in the healthcare system and the message were reinforced by provider referrals (lines 206 – 207); that it is an important finding of the RCT that there was 5% to 10% increase from baseline to follow-up in CBE and mammography rates among YBCS and relatives in both study arms (lines 214 – 215) – this is clinically relevant as it may translate into reduced morbidity and mortality (as noted in the Conclusion); that the findings suggest the intervention booklets and letters achieved higher acceptability and perceived usefulness among Black participants – as noted in the Introduction, there is an underutilisation of services that may detect breast cancer in this population, and as such, increasing access to such services, may reduce not only the burden of breast cancer morbidity and mortality in this population, but also the ethnicity based differences in breast cancer services and outcomes.

Discussion, page 11, lines 237 – 238: A strength of this study is the partnership between a state health department and a leading academic institution. Suggest that the authors add the study design as a further strength.

 Discussion, page 11, lines 241 – 247: The described limitations are well-discussed. Also suggest adding that there is a lack of generalizability to women older than 64, and to pregnant, imprisoned, institutionalised, or non-English speaking women. As breast cancer also occurs in men, suggest mentioning that results are not generalizable to men.

Materials and methods, page 11 – 12, lines 254 – 360: Overall, the description of materials and methods were clear, including: a clear description of the study design; settings and locations where data were collected; and outcome definitions. It would be useful to state whether or not any interim analyses were undertaken and if so, provide a brief summary of results and any outcomes affecting the trial, if relevant.

Materials and methods, page 11 – 12, lines 265-274:  The description of eligibility; inclusion and exclusion criteria is clear. As men are able to develop breast cancer, it would be advisable to state that men who had breast cancer, and their relatives, were excluded from this study, and why.

Materials and methods, page 12, line 296: While it is outlined that allocation was conducted via computer generated allocation algorithm, it would be useful to include who ran the computer generated allocation algorithm. Similarly, it would be useful to include who assigned participants to interventions. Please could the authors clarify if those assessing outcomes were blinded to assignment after intervention?

 Materials and methods, page 11 – 12, lines 304 - 309: The paper described that the follow-up survey was mailed to participants ~ 8 months after the intervention to allow sufficient time for pursuing the primary outcomes within the timeframe of the study. However, the described limitations include “allowing only 8 months to observe changes in outcomes”. It seems inconsistent to describe the 8 months as both sufficient time for pursuing outcomes, and a limitation for observing changes in outcomes. It is also a limitation that participants could not be blinded to the allocation, given the nature of the intervention.

Materials and methods, page 13 - 14, Line 327 - 360: The description of statistical analyses was clear and comprehensible. The methods were appropriate for the stated research aims, including Bonferroni corrections and sensitivity analyses.

Author Response

Please, see the attached document for a point-by-point response to comments of Reviewer 3.

Round 2

Reviewer 1 Report

Thank you for the authors tried to answer the questions and revised the manuscript. I have reviewed the author’s responses one by one including other reviewers. The authors have answered most questions that were raised by me. I only have a few minor concerns.

--Authors decided not to show 12 models in a table, because 1) self-efficacy and intention were not the primary outcomes of the trial, 2) the table is too busy, and 3) it would not provide additional information compared to what we have presented in the text. I suggested authors to provide relevant information in the supplementary files to avoid the misleading for readers.

--In this study, authors included cancer-free female relatives in the study because first- and second-degree relatives of YBCS have a 2.3 and 1.5 increased relative breast cancer risk respectively. In addition, they considered that YBCS are more likely to have an established relationship with the healthcare system and have cancer surveillance according to guidelines. However, cancer-free relatives may not have a routine source of care and an established relationship with a provider who would guide their screening practices. I totally agreed this opinion. Therefore, I recommend to discuss this issue in Discussion.

Author Response

Dear Reviewer,

we provided a supplementary table with regression analyses and we incorporated your suggestion below in the Discussion of the paper.

This manuscript is a resubmission of an earlier submission. The following is a list of the peer review reports and author responses from that submission.

Round 1

Reviewer 1 Report

I have read this paper now several times. It is unclear. I find the results misleading. For example the reporting of P values with out actual numbers in the abstract is not acceptable. The reader should have the ability to examine the numbers himself and decide if the effect is real.

This is true also in the results section specifically:

"Approximately 80% of YBCS and 70% of relatives reported CBE and mammograms consistent with NCCN guidelines at baseline. In the 8-month follow-up survey approximately 90% of YBCS and 82% of relatives reported having had CBE and mammograms consistent with NCCN guidelines, representing approximately10% increase from baseline."

however in Table 3 the actual numbers are different. For example mammography at baseline for the YBCS was 87% in both groups and increased slightly to 93 and 90%. this increase is not consistent with a 10% increase.

As there is no control group the actual increase in genetic testing in all groups is hard to evaluate, obviously as time passes more women will have a genetic test.